# Liquid-liquid phase separation in supercooled water from ultrafast heating of low-density amorphous ice

Katrin Amann-Winkel [1,2,10], Kyung Hwan Kim [3,10], Nicolas Giovambattista [4,5], Marjorie Ladd-Parada [1,9], Alexander Späh[1], Fivos Perakis [1], Harshad Pathak[1], Cheolhee Yang[3], Tobias Eklund[1], Thomas J. Lane [6], Seonju You [3], Sangmin Jeong[3], Jae Hyuk Lee [7], Intae Eom [7], Minseok Kim[7], Jaeku Park [7], Sae Hwan Chun [7], Peter H. Poole [8] ✉ & Anders Nilsson [1] ✉

Recent experiments continue to find evidence for a liquid-liquid phase transition (LLPT) in supercooled water, which would unify our understanding of the anomalous properties of liquid water and amorphous ice. These experiments are challenging because the proposed LLPT occurs under extreme metastable conditions where the liquid freezes to a crystal on a very short time scale. Here, we analyze models for the LLPT to show that coexistence of distinct high-density and low-density liquid phases may be observed by subjecting low-density amorphous (LDA) ice to ultrafast heating. We then describe experiments in which we heat LDA ice to near the predicted critical point of the LLPT by an ultrafast infrared laser pulse, following which we measure the structure factor using femtosecond x-ray laser pulses. Consistent with our predictions, we observe a LLPT occurring on a time scale < 100 ns and widely separated from ice formation, which begins at times >1 µs.

A growing body of experimental evidence supports the existence of a liquid-liquid phase transition (LLPT) in supercooled water, in which low-density liquid (LDL) and high-density liquid (HDL) phases become distinct below a critical temperature, $T_c$[1–8]. A LLPT has been experimentally observed in other systems such as phosphorous and sulfur, and evidence for a liquid-liquid critical point (LLCP) has recently been reported for the latter[9,10]. However, experiments testing for a LLPT in water are challenging because, unlike in phosphorous and sulfur, the proposed transition occurs under extreme metastable conditions for the liquid phase. Recent thermodynamic modeling predicts that the LLCP in water occurs at $T_c \simeq 220$ K and a critical pressure

$P_c \simeq 13$–72 MPa[11–13]. Under these conditions, crystalline ice begins to form in a bulk liquid sample on a time scale of 1 to 10 µs[8]. As a consequence, an experiment to observe the LLPT in supercooled water must be fast: It must first produce a liquid sample that is both supercooled and under pressure, and then measure its properties, all on a sub-µs time scale before crystallization intervenes.

A novel solution to this challenge was recently exploited to provide direct experimental evidence of the LLPT in supercooled water[8]. In this "pump-probe" approach, an ultrafast infrared (IR) laser pulse was used to heat a sample of high-density amorphous ice (HDA) to a point in the phase diagram at approximately 205 K and 300 MPa where

[1]Department of Physics, AlbaNova University Center, Stockholm University, SE-10691 Stockholm, Sweden. [2]Max Planck Institute for Polymer Research and Johannes Gutenberg University, 55128 Mainz, Germany. [3]Department of Chemistry, POSTECH, Pohang 37673, Republic of Korea. [4]Department of Physics, Brooklyn College of the City University of New York, Brooklyn, NY 11210, USA. [5]The Graduate Center of the City University of New York, New York, NY 10016, USA. [6]SLAC National Accelerator Laboratory, 2575 Sand Hill Road, Menlo Park, CA 94025, USA. [7]Pohang Accelerator Laboratory, Pohang, Gyeongbuk 37673, Republic of Korea. [8]Department of Physics, St. Francis Xavier University, Antigonish, NS B2G2W5, Canada. [9]Present address: Division of Glycoscience, School of Biotechnology, KTH Royal Institute of Technology, AlbaNova University Center, SE-10691 Stockholm, Sweden. [10]These authors contributed equally: Katrin Amann-Winkel, Kyung Hwan Kim. ✉e-mail: ppoole@stfx.ca; andersn@fysik.su.se

the system was in the HDL phase. Since the time required for heating was much less than the time required for sound propagation through the sample, the heating process was isochoric. After heating, the internal pressure of the sample relaxed to ambient pressure over a time scale of 10 ns to 100 µs, carrying the sample through the conditions predicted for the phase transition from HDL to the LDL phase. Hard x-ray laser pulses were used to characterize the structure of the sample as a function of time as the pressure decreased, and revealed two distinct transitions well separated in time, first from HDL to LDL, and then from LDL to ice[8]. Other recent pump-probe studies of amorphous ice are described in refs. [14,15].

Ultrafast isochoric heating is a valuable approach for studying the LLPT in water because it can move an amorphous solid sample to conditions relevant for studying the LLCP on a time scale (fs to ps) that is much shorter than can currently be achieved when cooling a bulk sample from the liquid phase (requiring 10–100 µs or more)[6]. In addition, the isochoric nature of the process provides access to thermodynamic pathways that are not commonly explored in experiments, and also means that the density of the starting material (HDA ice in ref. [8]) is an important control parameter that determines the pathway followed in the experiment, as illustrated in recent simulations[16,17].

Here, we examine the consequences of applying the same ultrafast isochoric heating procedure used in ref. [8] to low-density amorphous ice (LDA). From an analysis of thermodynamic models developed by Anisimov and coworkers[11–13], we first show that isochoric heating of LDA has the potential to move the sample to a state point much closer to the LLCP than is achieved by heating HDA. We then test this prediction in ultrafast laser heating experiments of LDA. Our experimental results are consistent with our predictions, and show that ultrafast heating of LDA provides a novel pathway to observe the rapid generation of coexistence of LDL and HDL phases within a single sample.

## Results

### Models of the binodal of the LDL−HDL phase transition

We plot in Fig. 1 several coexistence curves, or binodals, in the plane of density $\rho$ and temperature $T$ for the LDL-HDL phase transition estimated from thermodynamic modeling carried out by Anisimov and

coworkers[11–13]. All of these binodals are based on models which use as input the measured thermodynamic properties of real water over a large range of $T$ and pressure $P$. While there is considerable variation in the position of the binodal and the associated LLCP, all of these binodals have the curious property that the slope of the binodal line in the $\rho$-$T$ plane becomes negative on the low-density side as $T$ decreases. While this binodal shape is unusual in a single-component system, it is permitted by thermodynamics, and analogous behavior is often observed in $x$-$T$ phase diagrams for multi-component systems, where $x$ is the mole fraction of a particular component[18]. A notable implication of this binodal shape is that it means that it is possible to prepare a stable homogeneous state of the low-density phase at low $T$ that, upon isochoric heating, will enter the two-phase region inside the binodal where a homogeneous phase is either metastable or unstable at fixed volume. In this two-phase region, the previously homogeneous single-phase LDL system may decompose into two coexisting phases (LDL and HDL) having densities determined by the binodal. We note that the above thermodynamic reasoning neglects the fact that all these phases are metastable to crystalline ice in this region of the phase diagram. However, on a time scale shorter than that required for ice formation to begin, a LLPT between metastable phases may indeed be observed, as has been rigorously demonstrated in simulations of several water models[19–21].

Also shown in Fig. 1 is the density of LDA and HDA at $T = 80$ K and ambient pressure[22]. In particular, we notice that the density of LDA, 0.94 g cm$^{-3}$, has a value that lies between the minimum and maximum densities spanned by the binodals in two out of the five cases shown. It is therefore possible that LDA can be used as the initial state for an isochoric heating process that would take the system into the two-phase region inside the LDL-HDL binodal just below the critical temperature of the LLPT. Indeed, we see in Fig. 1 that the density of LDA at ambient pressure (0.94 g cm$^{-3}$) lies inside the range of estimates of the critical density $\rho_c$ of the LLCP, which vary from 0.928 to 0.988 g cm$^{-3}$.

The binodals shown in Fig. 1 are the most recent estimates (since 2012) of the LDL-HDL binodal in the $\rho$-$T$ plane. There have been a number of estimates of $T_c$ and $P_c$ of the LLCP, both from experiments and thermodynamic modeling[2–5,7,11–13]. The values of $T_c$ and $P_c$ for the models corresponding to each binodal in Fig. 1 fall respectively in the ranges 215–227 K and 13–72 MPa. These values are consistent with the range of estimates reported in other work. However, the most recent estimates for $\rho_c$ and the binodal curve are restricted to the cases shown in Fig. 1. There is significant variation in $\rho_c$ in Fig. 1, depending on the choice of experimental data used to fit the model parameters. The two most recent binodal curves in Fig. 1 (black and red) do not straddle the density of LDA and lie at higher density. However, ref. [13] shows that when the most recent data for the isothermal compressibility[7] are included in the fit, the estimate of $\rho_c$ decreases, from that on the red curve, to that on the black curve. Given the variation in the estimates for $\rho_c$ and for the location of the binodal curve, an isochoric heating experiment starting from LDA would provide a direct way to test the different predictions shown in Fig. 1.

Figure 2a is a simplified version of Fig. 1 that focusses on the behavior that would occur based on the blue binodal from Fig. 1. Isochoric heating of HDA (red arrow) results in a pure HDL system at 205 K, as explored along the pathway in the experiments of ref. [8]. Isochoric heating of LDA to 200 K (black arrow) brings the system to a point inside the binodal where the equilibrium state (if the crystalline phase is ignored) is a phase separated system made up of coexisting regions of LDL and HDL. The state created by isochoric heating is much closer to the LLCP when starting from LDA than from HDA. Our analysis therefore suggests that isochoric heating of LDA provides a novel pathway that might give direct access to state points very close to the LLCP and its associated critical fluctuations, and at which phase separation and coexistence of LDL and HDL could be directly observed.

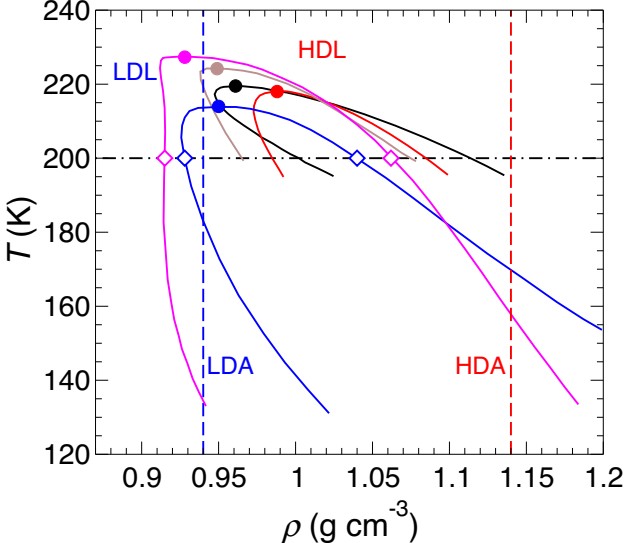

**Fig. 1 | Comparison of estimates for the LDL−HDL binodal.** Solid lines are binodals predicted from thermodynamic modeling of experimental data in ref. [11] (blue and brown), ref. [12] (magenta), and ref. [13] (black and red), with corresponding critical points marked by filled circles. Diamonds locate the coexistence densities of LDL and HDL at 200 K as determined by the magenta and blue binodals. Vertical dashed lines are isochores at the density of LDA (blue) and HDA (red) at $T = 80$ K and ambient pressure[22]. The dot-dashed line is an isotherm at 200 K.

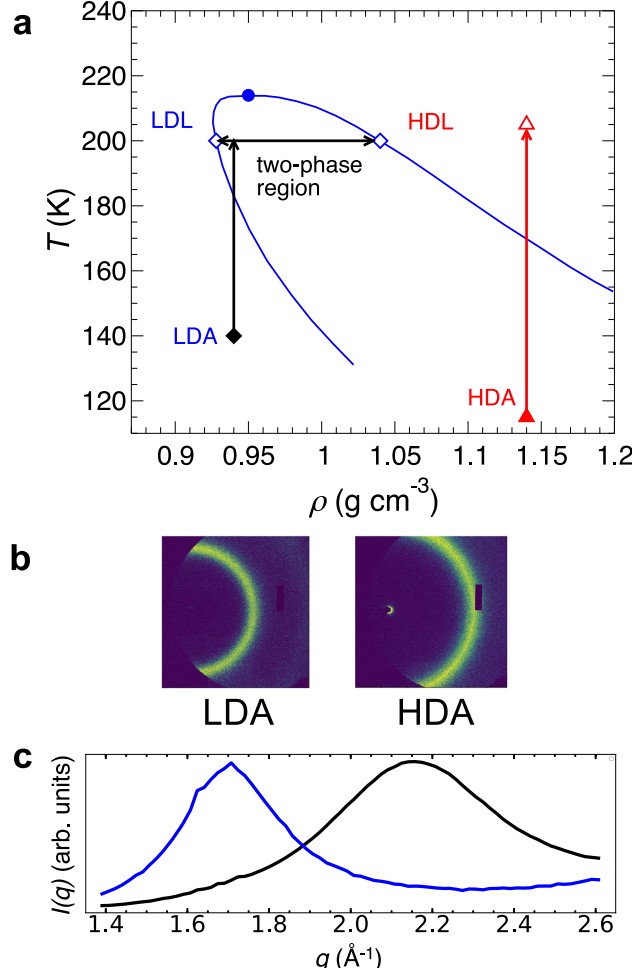

**Fig. 2 | Isochoric heating of LDA to access the LDL–HDL coexistence region.**
**a** The blue curve is the binodal predicted from thermodynamic modeling in ref. [11], with the corresponding critical point marked by the filled circle. The red arrow corresponds to an isochoric heating pathway, of the kind explored in ref. [8], in which HDA (filled triangle) is heated to HDL (open triangle). The vertical black arrow illustrates the isochoric heating pathway explored here, in which LDA (filled diamond) is heated to a liquid state inside the two-phase region of the binodal. This state then phase separates into LDL and HDL components (horizontal arrows). Open diamonds locate the coexistence densities of LDL and HDL at 200 K as determined by the binodal. **b, c** Characteristic x-ray scattering intensities for LDA and HDA as observed (**b**) on the 2D experimental detector and (**c**) as $I(q)$ scattering intensity curves as a function of wave number $q$. The LDA x-ray data shown in (**b, c**) is for the LDA sample prior to IR heating. The HDA x-ray data in (**b, c**) is for a HDA sample as reported in ref. [8]. $I(q)$ for LDA and HDA is nearly identical to that for LDL and HDL, respectively[8, 30].

## Experimental pump-probe measurements

The experiments were conducted at the XSS-FXS beamline of PAL-XFEL[23,24]. LDA samples were mounted in a cryostat inside a vacuum chamber to allow for pump-probe measurements in a transmission geometry, using a combination of ultrafast IR and x-ray laser pulses[8]. See "Methods" and Supplementary Note 1 for details on sample preparation and data collection. The samples were between 40 and 80 µm thick and maintained at 140 K for several hours prior to heating. Each sample spot was first pumped by a 100 fs IR laser pulse of wavelength 2 µm, corresponding to the excitation of the combination mode of O–H stretch and H–O–H bending modes. The IR pulse raised the sample temperature from 140 K to 200 K within ≈20 ps. As in the experiments of ref. [8], the time scale of the heating process is much shorter than the time required for the density to respond to the change in $T$, which is

limited by the speed of sound in the sample[25]. As a result, the heating process is isochoric. The final temperature reached after the IR pulse (200 K) was estimated based on analysis of Bragg reflections after crystallization[26,27]; see "Methods" for details. After the IR heating pulse, the sample was probed with an x-ray pulse of 9.7 keV at various time delays $\Delta t$ after the IR pulse ended, from 8.4 ns to 10 µs. The temperature of the laser-heated region of the sample, 200 K, remains effectively constant for all $\Delta t$ studied here because re-thermalization with the surrounding material at 140 K occurs on a time scale greater than 100 µs[8].

Figure 3a shows the x-ray scattering intensity $I(q)$ as a function of wave number $q$ for several $\Delta t$. We focus on the q range of the first sharp diffraction peak, or "pre-peak", in $I(q)$ for 1.4 Å⁻¹ < q < 2.6 Å⁻¹. In supercooled liquid and amorphous solid water, the shape and position of this peak is highly sensitive to the presence of low-density or high-density components in the sample[28,29]. The bottom panel in Fig. 3a shows $I(q)$ for the sample prior to IR heating, and exhibits the peak shape characteristic of pure LDA at 140 K with a maximum at $q = 1.7$ Å⁻¹; see Fig. 2b, c and refs. [8,30]. At $\Delta t = 8.4$ ns, we observe a decrease in intensity of the peak at 1.7 Å⁻¹ and the emergence of a lower and broader peak with a maximum at 2.15 Å⁻¹. This new feature has the same shape and position of the characteristic peak observed for both HDA and HDL, shown here in Fig. 2b,c and in refs. [8,30]. The change in the scattering pattern persists at all delays up to 10 µs when crystallization starts to occur, as shown by the appearance of sharp Bragg peaks corresponding to the structure of stacking-disordered ice, $I_{sd}$[8].

Prior to crystallization, the sample has a temperature of 200 K, nearly the same as the temperature (205 K) of the pumped HDA sample studied in ref. [8]. As discussed in detail in ref. [8], the high-density and low-density components of the sample in this temperature range are well above their glass transition temperatures and are liquids on the time scale of our observations. Liquid state relaxation for HDL is achieved in less than 10 ns, and for LDL on a time scale of the order of 100 ns. On this basis, the two components observed here in $I(q)$ following the IR pulse can be interpreted as arising from distinct regions of LDL and HDL occurring in the sample. The changes in $I(q)$ can be seen more in detail in Fig. 3b, which shows the difference $\Delta I(q)$ between the unpumped and pumped sample at different $\Delta t$. The peak associated with HDL appears throughout the range of $\Delta t$ prior to crystallization. As shown in Fig. 3b, the experimental $\Delta I(q)$ curve can be well fitted by a weighted sum of components from the scattering patterns of pure HDA, LDA and $I_{sd}$. The fitting procedure used to estimate these contributions is the same as that described in ref. [8]. Since $I(q)$ for LDA and HDA is nearly identical to that for LDL and HDL respectively[8,30], this fitting procedure estimates the fraction of the sample in the LDL and HDL phases.

We emphasize that the time evolution of $I(q)$ in Fig. 3 is consistent with the occurrence of a LLPT, and is not consistent with other scenarios (such as the "singularity-free scenario"[5]) in which the sample is a homogeneous mixture of local, nm-scale LDL-like and HDL-like domains. If the system were homogeneous, additional scattering due to pervasive interfaces between such small LDL-like and HDL-like regions would cause the two distinct peaks we observe in $I(q)$ to blend into one, the peak of which would shift in q as the transformation progressed, from $q_{LDL} = 1.7$ Å⁻¹ to $q_{HDL} = 2.15$ Å⁻¹; see Fig. 2C. Instead, we observe a superposition of two distinct peaks, with maxima at q values that are invariant with time, as expected in phase separation of domains of HDL and LDL that are macroscopic in size, so that contributions from domain interfaces is negligible[29–31]. This reasoning is described in detail in refs. [8,32].

Figure 4 shows the mass fraction $x$ of the HDL, LDL and crystalline ice components present in the sample as a function of $\Delta t$, as obtained from the fits of the $\Delta I(q)$ curves in Fig. 3b. We observe the rapid appearance of HDL with $x_H \approx 0.10$ at $\Delta t = 8.4$ ns, at the expense of the LDL component $x_L$. For at least two orders of magnitude in time, from 0.01 to 1 µs, the sample is composed solely of LDL and HDL

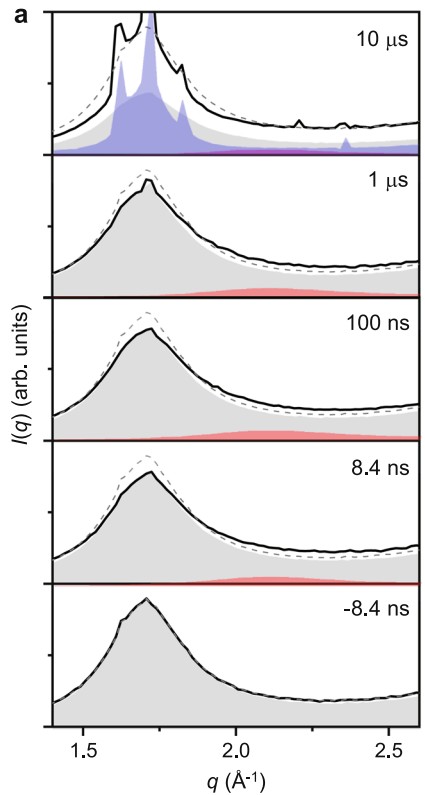
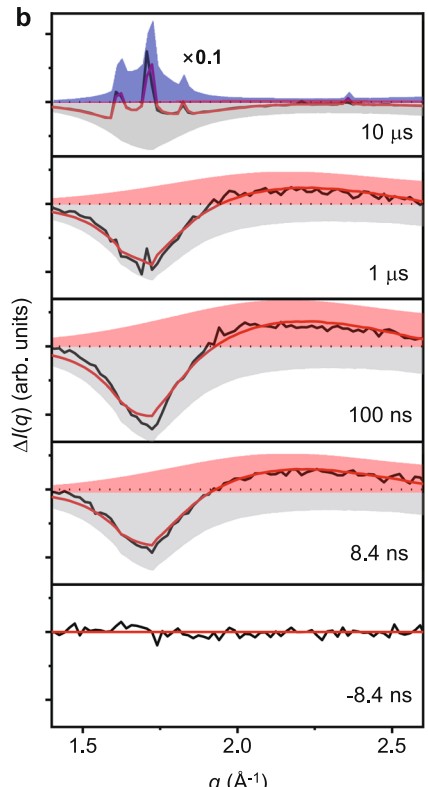

**Fig. 3 | Structure factor before and after heating LDA. a** Experimental x-ray scattering intensities, $I(q)$, of LDA samples measured before (gray dashed line) and after the laser excitation (black solid line). Data obtained at IR pump/x-ray probe delay times of $\Delta t = -8.4$ ns to 10 µs are shown. Each curve is obtained from the average of $I(q)$ over three independent x-ray shots. The contributions from LDA, HDA, and crystalline ice are indicated as gray, red, and blue shaded areas,

respectively. We note that $I(q)$ for LDA and HDA is nearly identical to that for LDL and HDL respectively[8,30]. **b** Difference scattering intensities $\Delta I(q)$ (black solid lines), at different delay times $\Delta t$, obtained by subtracting the pre-excitation $I(q)$ from the post-excitation $I(q)$ shown in (**a**). The differences are fit (red solid line) as a combination of depletion of LDA (gray shaded area), formation of HDA (red shaded area), and formation of crystalline ice (blue shaded area).

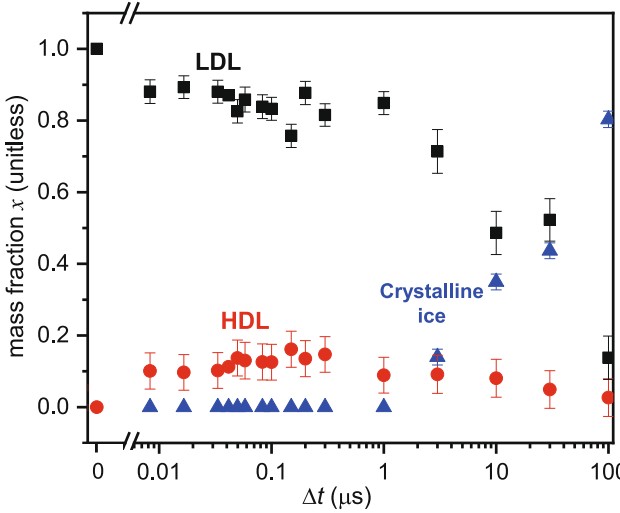

**Fig. 4 | Time evolution of LDL, HDL and ice components after heating.** Mass fraction of LDL (black squares), HDL (red circles), and crystalline ice (blue triangles) as a function of the pump-probe delay time $\Delta t$. Each data point is obtained by fitting our $\Delta I(q)$ curves, averaged over three x-ray shots, to a sum of contributions due to LDA, HDA and crystalline ice, using the same procedure as described in ref. [8]. The error bars show the standard error for each data point determined by propagating the error associated with the averaged $\Delta I(q)$ curves.

components, with the HDL component comprising between 0.1 and 0.15 of the sample. At $\Delta t = 3$ µs, crystalline ice starts to appear and eventually consumes the entire sample at the expense of both the HDL and LDL components.

The appearance of the HDL component and the observed values of $x_H$ can be understood in terms of the path illustrated in Fig. 2a. Isochoric heating of the LDA sample drives the system to a point inside the LDL–HDL binodal at sufficiently high $T$ so that the sample is now a liquid. Under these conditions, the stable state of the system prior to ice formation is a two-phase coexistence of LDL and HDL, and so a HDL component rapidly appears in the $I(q)$ signal. Since the starting density of the sample is much closer to the LDL side of the LDL–HDL binodal, the HDL phase is expected to be at best a minority component of the sample, which is what we observe. In Fig. 1, two of the model binodals (the magenta and blue curves) straddle the density of LDA, $\rho_{LDA} = 0.94$ g cm$^{-3}$. We find the coexistence densities of LDL and HDL predicted by these binodals at 200 K, denoted as $\rho_L$ and $\rho_H$ respectively, and then estimate the equilibrium mass fraction $x_H$ of HDL in the sample at 200 K when the overall sample density is $\rho_{LDA}$, using the "lever rule" for the mass fraction expressed in terms of the various densities: $x_H = (\rho_{LDA}^{-1} - \rho_L^{-1})/(\rho_H^{-1} - \rho_L^{-1})$[18]. For the magenta curve we find $x_H = 0.19$ and for the blue curve we find $x_H = 0.12$. These values straddle the largest value of $x_H = 0.15$ observed in Fig. 4, and thus demonstrate that these model binodals may provide viable descriptions of the LLPT.

The variation of the data for $x_H$ with $\Delta t$ is comparable with the size of the error shown in Fig. 4, and so our results do not unambiguously reveal the time dependence of $x_H$ within the time window where no ice is observed. Nonetheless, the trend in the data is that $x_H$ seems to

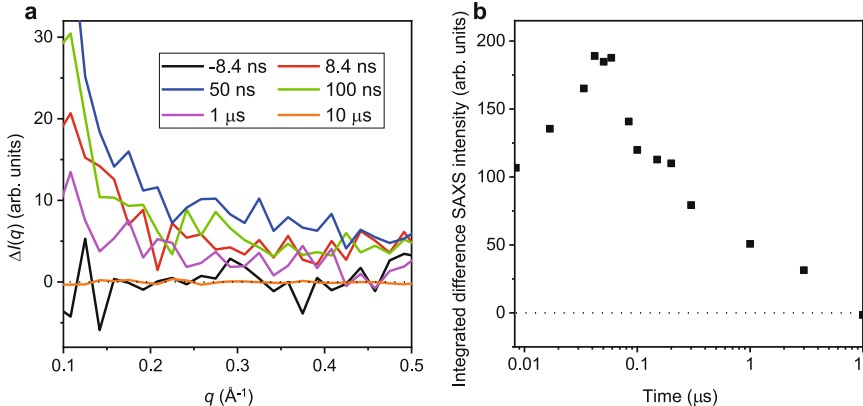

**Fig. 5 | Small-angle x-ray scattering results. a** The difference between the unpumped and pumped scattering curves at various time delays in the SAXS region. **b** Time-dependent integrated difference SAXS intensity from $q = 0.1$ to 0.3 Å⁻¹.

saturate to its largest values on a time scale of approximately 50 to 300 ns. The time scale for the full development of a two-phase LDL–HDL system will be limited by the slower relaxation time of the LDL phase, which is predicted to have a relaxation time on the order of 100 ns, a time scale consistent with the largest observed values of $x_H$. On longer time scales the appearance and growth of the crystalline ice component causes both $x_H$ and $x_L$ to decrease.

We find additional evidence of a LLPT from the small-angle x-ray scattering (SAXS) intensity. Figure 3 shows data in the wide-angle (WAXS) region where the lowest $q$ is limited to 1.35 Å⁻¹. We have also made measurements in the $q$ range down to 0.1 Å⁻¹ where we find enhanced scattering, as shown in Fig. 5(a). Figure 5b shows the time dependence of the integrated difference of the SAXS intensity from $q = 0.1$ to 0.3 Å⁻¹ between the pumped and unpumped data. The enhanced SAXS scattering, which is increasing for delay times up to approximately 60 ns, is consistent with the appearance of nm-scale LDL and HDL domains. These small domains may arise due to rapid nucleation at multiple sites throughout the sample, or a spinodal decomposition process. On a time scale between 60 ns and several hundred ns, the SAXS intensity decreases, while the intensity of the signal from our WAXS measurements remains approximately constant, as shown in Fig. 4, consistent with the consolidation and growth of macroscopic HDL and LDL domains from the nm-scale domains appearing at earlier times. This is the behavior expected in a first-order phase separation process.

## Discussion

In summary, our experimental results support the possibility, predicted by thermodynamic modeling, that isochoric heating of LDA drives the sample inside the region of the coexistence curve along the path illustrated in Fig. 2. While the magnitude of the HDL signal in Fig. 4 is small, our analysis of the thermodynamic models summarized in Fig. 1 shows that the HDL signal, if it appears at all, should be small, and on the order of the value of $x_H$ that is actually observed. Indeed, our results suggest that the LDL–HDL binodal curve straddles the 0.94 g cm⁻³ isochore in the vicinity of 200 K. This result provides a new experimental constraint to guide the development of thermodynamic and simulation models.

It is worth noting that we expect there to be a temperature gradient across the sample along the IR beam axis. Ref. [8] estimates that $T$ decreases by approximately 20 K across the thickness of the sample, if the absorption coefficient of LDA is assumed to be the same as ice. If a gradient occurs, the colder side of the sample may not undergo phase separation because it remains outside the binodal curve of the LDL–HDL transition. In this case, our results for the HDL fraction represent a lower bound, and could in fact be higher.

We also point out that although the HDL phase is observed as the minority component in the process explored here, it is difficult to account for the presence of the HDL signal in $I(q)$ if the LLPT is not occurring in our sample. The HDL peak in $I(q)$ is well separated from the LDL peak, and is straightforward to detect because it occurs close to where $I(q)$ for LDL has a minimum, as highlighted by the $\Delta I(q)$ plots in Fig. 3B. Furthermore, during heating and subsequent decompression starting from the pure LDA sample, the overall sample density should never exceed the initial density of 0.94 g cm⁻³. Nonetheless, a broad non-crystalline peak appears in $I(q)$ associated with a distinctly denser form of water, HDL. The LLPT is the only mechanism currently known that could produce this effect, where macroscopic regions of a high-density non-crystalline component spontaneously appear in a system having an overall lower density.

We note that our results arise by exploiting several unusual and independent factors: (i) the ability to explore isochoric heating pathways; (ii) the peculiar shape predicted for the HDL-LDL binodal; and (iii) the close correspondence of the density of LDA to the estimated critical density of the LLCP. In the present study, these factors converge to provide a new opportunity to access the region of the water phase diagram where the LLPT may be observed, following a single-step pathway that leads directly to conditions close to the LLCP. Future work that builds on the procedure described here therefore has the potential to test for the critical fluctuations expected in the vicinity of the LLCP, recently quantified in simulations[20].

## Methods

### Sample preparation

The samples were prepared ex-situ in Stockholm through the well-established method of pressure amorphization, in which ultrapure hexagonal ice is first compressed to HDA and subsequently decompressed to LDA at 140 K[29,33–35]. A sheet of Cu (of thickness 100 µm) with a grid of circular holes (each of diameter 1.5 mm) was used as a sample holder. Water was loaded and transformed into amorphous ice inside the holes of the Cu-grid, producing a free-standing film of amorphous ice within each hole, suitable for pump-probe measurements under vacuum (less than 0.1 Pa). The samples do not need to be covered by windows, and the Cu-grid can be clamped directly to the sample holder[8].

First, hexagonal ice is formed in the Cu-grid and then compressed inside a piston cylinder apparatus to form unannealed HDA at 100 K and 1.6 GPa[33]. We then follow the established procedure to form LDA through decompression of temperature-annealed HDA at 140 K[29,34,35]. The Cu-grid containing LDA is quench recovered, stored, and shipped to the PAL facility at liquid nitrogen temperatures. Throughout the process, the ice in the Cu-grid is protected by a 0.2 mm thick indium

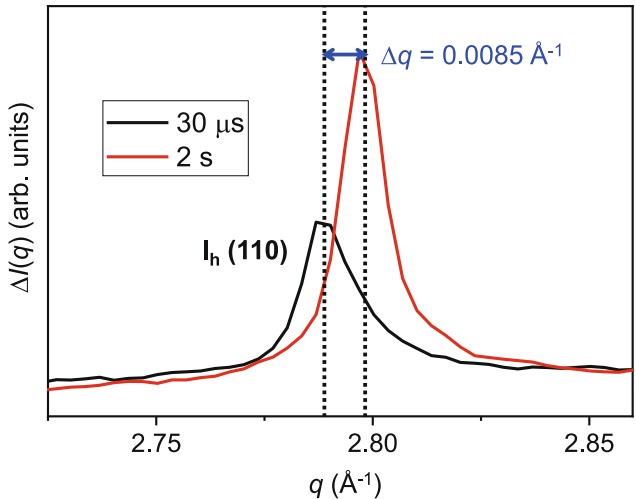

**Fig. 6 | Bragg peak shift in ice following laser heating.** Ice $I_h$ (110) peak positions measured at 30 μs (black) and 2 s (red) after laser excitation of the sample. The shown peak shift of 0.0085 Å$^{-1}$ corresponds to a temperature difference of approximately 90 K in ice $I_h$. As described in the text, this temperature change implies that the temperature of the LDA sample was increased by 60 K by the IR laser pulse, prior to crystallizing.

sheet, to avoid condensation, and is removed just prior to the experiment. After removal of the indium cover, the thickness of the amorphous ice sheet is between 40 and 80 μm for the samples chosen for examination; see Supplementary Note 1. The sample is then cold-loaded into the sample holder inside a liquid nitrogen bath and transferred to the vacuum chamber.

Prior to heating the sample with the IR laser pulse, the LDA sample is maintained at a temperature of 140 K for several hours. The glass transition temperature for LDA is estimated to be $T_g = 136$ K, and so our sample at 140 K may be in a highly viscous LDL state[35]. Nonetheless, to avoid confusion, in the main text we refer to the sample at 140 K as LDA, to distinguish the state of the sample prior to IR heating from the much more fluid LDL state at 200 K after heating.

## Data collection at PAL-XFEL
The experiment was performed at the XSS-FXS beamline of PAL-XFEL. The detailed experimental procedures are described in our previous study, ref. [8]. Briefly, an infrared-pump x-ray-probe scheme has been applied to obtain time-dependent single-shot x-ray scattering patterns of amorphous ice samples. Femtosecond IR laser pulses at 2000 nm wavelength with 250 μJ/pulse were focused to a spot of 70 μm (FWHM) diameter, where the laser beam was overlapped with the x-ray beam with a crossing angle of 20°. 50 fs x-ray pulses with a mean energy of 9.7 keV and an energy bandwidth of 0.3% ($\Delta E/E$) were focused on a spot of 19 μm × 32 μm at the sample position. A wide $q$ range (0.1–3.2 Å$^{-1}$) was covered by using a large-area CCD detector (MX225-HS, Rayonix) at a distance of 250 mm. An ultrafast temperature-jump ($T$-jump) of approximately 60 K (see below) within the sample was introduced by a single IR pump pulse and the time-dependent change of the sample was probed by a single x-ray pulse. To provide a fresh sample for each individual measurement, the sample holder was moved to a fresh position before each pump-probe shot. Laser-off images were acquired as a reference and used for obtaining the time-resolved difference x-ray scattering patterns. To cover the entire process after the $T$-jump, the scattering images were measured at the following time delays $\Delta t$: −8.4, 8.4, 16.8, 33.6, 42, 50.4, 58.8, 84, 100, 150, 200, 300 ns, 1, 3, and 10 μs. The dataset was collected using a sample base temperature of 140 K and a laser fluence of 250 μJ/pulse.

## Temperature-jump estimation
Since the exact absorption cross-section and the heat capacity of LDA at the conditions used in the current experiment are not known, the temperature increase in the sample upon IR illumination must be estimated by other means. We estimate the $T$-jump by measuring the Bragg peak shift in $I(q)$ of crystalline hexagonal ice generated after the laser excitation of the LDA sample. In Fig. 6, the ice $I_h$ (110) peak positions measured at 30 μs and 2 s after IR illumination are compared. At $\Delta t = 30$μs we conclude that the increased pressure inside the probed volume induced by the IR pulse is fully released because there is no significant peak shift between 10 μs and 100 μs. However, the heat has not yet dissipated to the surroundings due to the low thermal diffusivity of ice. Thus, the probed volume maintains an approximately constant and elevated temperature on a time scale of 100 μs. On a time scale of 2 s, the temperature in the probed volume is expected to be fully equilibrated with the temperature of the cryostat, 140 K, and consistent with this we observe no significant peak shift between 2 s and 5 s. Therefore the temperature change induced by the IR pulse can be estimated from the peak shift shown in Fig. 6, by comparing the peak position at 30 μs, when the sample temperature is still elevated, and the peak position at 2 s, when the sample temperature has returned to 140 K. The observed shift is $\Delta q \approx 0.0085$ Å$^{-1}$. Using the known temperature-dependent change of the lattice constant of ice $I_h$[26,27] we estimated a temperature change of approximately 90 K from the peak shift in Fig. 6.

This change in $T$ overestimates the $T$-jump induced by the IR pulse on the LDA sample because it includes the effect of the latent heat released when the sample crystallizes (333 kJ/kg at 273 K), and also does not account for the difference in heat capacity $C_P$ between liquid water and ice $I_h$ (approximately 3 kJ kg$^{-1}$ K$^{-1}$). We assume that we are close to the LLCP after the IR pulse, and so anticipate no significant heat release from the LLPT itself. With these corrections applied, the $T$-jump of the LDA sample after 250 μJ of laser excitation is estimated to be approximately 60 K. Since the front of the sample absorbs more energy than the back, a temperature gradient is likely generated. The $T$-jump estimated here is an average value within the probed volume. As shown in Supplementary Note 1, we estimate the variation of the temperature in a sample of thickness 60 μm to be 60 ± 10 K.

## Data availability
The authors declare that all data supporting the findings of this study are available within the paper and its supplementary information files. Source data are provided with this paper.

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

## Acknowledgements

This work has been supported by a European Research Council Advanced Grant under project 407 667205 (A.N.) and a Swedish National Research Council Grant 2013-8823 (A.N.). K.A.-W. acknowledges funding by the Ragnar Söderbergs Stiftelse (S14/20). T.J.L. was supported by U.S. Department of Energy contract DE-AC02-76SF00515. This work is also supported by the National Research Foundation of Korea (NRF) grant funded by the Korea government (MSIT) (2019R1C1C1006643 and NRF-2020R1A5A1019141) (K.H.K., C.Y., S.Y. and S.J.). The experiments were performed at beamline XSS of PAL-XFEL (proposals 2018-1st-XSS-009, 2018-2nd-XSS-006, and 2019-1st-XSS-008) funded by the Korea government (MSIT). N.G. thanks the NSF CREST Center for Interface Design and Engineered Assembly of Low Dimensional systems (IDEALS), NSF grant number HRD-1547380, for support. P.H.P. is supported by the Natural Sciences and Engineering Research Council of Canada (NSERC), Grant No. RGPIN-2017-04512.

## Author contributions

A.N. K.A.-W. K.H.K. and P.H.P. designed and supervised the study. K.A.-W. designed the sample preparation. K.A.-W and A.S designed the sample holder. K.A.-W., T.E., A.S. and M.L.P. prepared the ice samples. K.H.K., K.A.-W., A.N., A.S., F.P. and H.P. designed the experimental setup, chamber and laser geometry. K.H.K., K.A.-W., A.N., A.S., F.P., H.P., M.L.P., C.Y., T.E. T.J.L, S.Y., S.J., J.L., I.E., M.K., J.P., and S.H.C. performed the x-ray experiments. N.G. and P.H.P. carried out the thermodynamic analysis depicted in Figs. 1 and 2a. K.H.K., A.S., C.Y., S.Y. and S.J. analyzed the data. P.H.P., A.N., K.H.K., K.A.-W. and M.L.P. wrote the manuscript.

## Competing interests

The authors declare no competing interests.
