## [Peer Review File · Nature Communications]

Liquid-liquid phase separation in supercooled water from ultrafast heating of low-density amorphous iceREVIEWER COMMENTS

Reviewer #1 (Remarks to the Author):

In the submitted manuscript, the authors investigate the structural changes in a low-density amorphous (LDA) water sample that is rapidly heated to ~200 K, transforming it to a low-density liquid (LDL). The resulting structural changes are investigated as a function of time with x-ray scattering techniques. The main experimental result is that a (small) signal arises due to the appearance of a high-density liquid (HDL) component within the low-density phase before the sample eventually crystallizes. The authors provide good evidence that the appearance of the HDL signal shows that liquid-liquid phase separation has occurred in the sample. The results presented here are closely related to an earlier publication in which this group investigated structural changes versus time after a high-density amorphous (HDA) water sample was rapidly heated. However, the results presented here are distinct from the earlier publication and of sufficient interest to warrant publication in Nature Communications. Because the possible existence of a liquid-liquid transition in supercooled water and, if so, the location of its associated critical point are issues of significant interest, information regarding the location of critical point is a significant result. Therefore, I recommend publication.

Reviewer #2 (Remarks to the Author):

Report

The authors present the results of a pump (100-fs infrared laser) and probe (X-ray free electron laser) experiments of supercooled water. Analyzing the temporal evolution of scattering intensity profiles after the ultrafast isochoric heating of low-density amorphous ice (LDA) induced by the infrared laser pump, the authors claim to observe the coexistence of low-density liquid (LDL) water and high-density liquid (HDL) water before the formation of ice.

This work is a follow-up of the authors' previous work (Ref. 8, Science 370, 978-982 (2020)), in which high-density amorphous ice (HDA) was subject to isochoric heating using the same procedure. This time, by using LDA, the authors claim, based on the binodal curves estimated from thermodynamic modeling (Ref. 11-13), that one can explore the region of the phase diagram which is closer to the putative liquid-liquid critical point, which is in "no man's land." This work, together with Ref. 8, demonstrates that the authors' approach is useful for exploring the phase diagram within the no man's land.

Compared to the careful work on HDA (Ref. 8) conducted by the same group, the manuscript lacks the detailed examination of experimental results/artifacts that might potentially influence their argument; this also makes me feel that this work is rather incremental research to the existing work. For example, in their previous work (Ref. 8), the authors showed the existence of thickness inhomogeneity (from 15 μm to ~80 μm) and discussed how the temperature gradient could influence the results; in this work, only the averaged thickness of 80 μm is reported. Is the thickness of LDA homogeneous compared to the HDA?

Regarding the temperature distribution across the sample, which is one of the biggest uncertainties in this approach, the authors discuss that "Ref [8] estimates that T decreases by approximately 20 K across the thickness of the sample...". However, that estimation in Ref. 8 is for 45 μm thickness (Fig. S8 in Ref. 8), while the thickness used in this work is 80 μm , leading to a much larger temperature gradient. As the authors discuss, the current result will be able to be used to set the lower bound of HDL fraction. However, I believe the authors can better discuss this issue, together with a possible thickness inhomogeneity. Related to this, in their previous work, the temperature increase in HDL using the identical laser (2000 nm wavelength, 250 $\mu\text{J}/\text{pulse}$ with a spot size of 70 μm) was 90 K, compared to 60 K in this work. What caused this difference?

The temporal evolution of HDL and LDL, as well as ice, is discussed using Fig. 4. Here, the guide to

the eye is introduced smartly but is not necessarily justified in my view. From 8.4 ns to ~ 150 ns, the guide to the eye indicates a slight increase in the HDL fraction and a slight decrease in the LDL fraction, but they may be interpreted as constant within the uncertainty. The definition of uncertainty is not shown, and the number of XFEL shots used for obtaining single data points is not described either. This lack of description and justification significantly weakens the discussion from L.181 to L.216. For example, the statement “The gradual decrease in x_H occurring for $\Delta t > 300$ ns is consistent...” at line 214 is solely based on the result that a single data point at $\Delta t = 1 \mu\text{s}$ is slightly smaller than that at $\Delta t = 300$ ns before the formation of ice is observed. No gradual decrease can be discussed from the current dataset.

The authors exclude the possibility of the formation of nm-scale LDL-like & HDL-like domains, while their SAXS results indicate the appearance of nm-scale nucleation or spinodal decomposition process. Even though the authors describe the growth of macroscopic HDL and LDL observed in WAXS follows the appearance of SAXS intensity, in my view, they occur almost simultaneously.

L.351-363: the authors discuss the scattering intensity from a two-component system. I understand the authors' intention here, but eqs.1 and 2 are not necessarily correct, particularly for the X-rays from XFEL. Instead of $(S_1+S_2)^2$, the intensity is the absolute square of summed scattering amplitude, so the cross-term can be almost cancelled even if the interface cannot be neglected. As I said, I understand the authors' intent, but considering the readership of Nature Communications, this paragraph needs to be revised.

Despite the several concerns raised above, I mostly support the authors' argument, which needs more justifiable discussion. However, I am not sure this work deserves publication in Nature Communications, considering rather incremental progress compared to their previous work.

In the following, we reproduce the comments of each reviewer in boldface, followed by our response.

Reviewer: 1

In the submitted manuscript, the authors investigate the structural changes in a low-density amorphous (LDA) water sample that is rapidly heated to ~200 K, transforming it to a low-density liquid (LDL). The resulting structural changes are investigated as a function of time with x-ray scattering techniques. The main experimental result is that a (small) signal arises due to the appearance of a high-density liquid (HDL) component within the low-density phase before the sample eventually crystallizes. The authors provide good evidence that the appearance of the HDL signal shows that liquid-liquid phase separation has occurred in the sample. The results presented here are closely related to an earlier publication in which this group investigated structural changes versus time after a high-density amorphous (HDA) water sample was rapidly heated. However, the results presented here are distinct from the earlier publication and of sufficient interest to warrant publication in Nature Communications. Because the possible existence of a liquid-liquid transition in supercooled water and, if so, the location of its associated critical point are issues of significant interest, information regarding the location of critical point is a significant result. Therefore, I recommend publication.

We thank the reviewer for their careful reading of our work. We are pleased with the Reviewer's positive and supportive comments, and note that the Reviewer does not request any revisions of our manuscript.

Reviewer: 2

We thank the Reviewer for taking the time to consider our results and provide a thoughtful review that raises several interesting points. As documented below, addressing these questions has helped us to develop a stronger and clearer manuscript.

The authors present the results of a pump (100-fs infrared laser) and probe (X-ray free electron laser) experiments of supercooled water. Analyzing the temporal evolution of scattering intensity profiles after the ultrafast isochoric heating of low-density amorphous ice (LDA) induced by the infrared laser pump, the authors claim to observe the coexistence of low-density liquid (LDL) water and high-density liquid (HDL) water before the formation of ice.

This work is a follow-up of the authors' previous work (Ref. 8, Science 370, 978-982 (2020)), in which high-density amorphous ice (HDA) was subject to isochoric heating using the same procedure. This time, by using LDA, the authors claim, based on the binodal curves estimated from thermodynamic modeling (Ref. 11-13), that one can explore the region of the phase diagram which is closer to the putative liquid-liquid critical point, which is in "no man's land." This work, together with Ref. 8, demonstrates that the authors' approach is useful for exploring the phase diagram within the no man's land.

Compared to the careful work on HDA (Ref. 8) conducted by the same group, the manuscript lacks the detailed examination of experimental results/artifacts that might potentially influence their argument; this also makes me feel that this work is rather incremental research to the existing work. For example, in their previous work (Ref. 8), the authors showed the existence of thickness inhomogeneity (from 15 μm to $\sim 80 \mu\text{m}$) and discussed how the temperature gradient could influence the results; in this work, only the averaged thickness of 80 μm is reported. Is the thickness of LDA homogeneous compared to the HDA?

We have added a description of the thickness variation of the samples in the Supplementary Information file. As described there, the samples used in our experiments are chosen from those having a thickness in the range between 40 and 80 μm . Our original manuscript stated that the average thickness of our samples is 80 μm . This value is correct for the average thickness of all our LDA samples, but is not correct as a description of the average thickness of the samples selected for the pump-probe experiments. We have corrected the main manuscript text accordingly. We thank the Reviewer for asking about this issue, which has allowed us to clarify our description and identify the error in the main text.

The Supplementary Information text, including the figure, reads:

The thickness of each individual sample spot is calibrated by comparing the integrated scattering intensity around the first peak of $S(q)$ with the value measured from a sample of known thickness. The thickness distribution is shown in Figure S1(A). The results in the main text were obtained by restricting the analysis to samples of thickness between 40 and 80 μm .

Figure S1. (A) Thickness distribution of the sample spots that is used for the measurement. The thickness range used for the main data is indicated in blue.

We have also updated the text of the manuscript in both the Results (lines 132-133) and Methods sections (lines 273-274) to state that we used samples with a thickness in the 40 to 80 μm range.

Regarding the temperature distribution across the sample, which is one of the biggest uncertainties in this approach, the authors discuss that “Ref [8] estimates that T decreases by approximately 20 K across the thickness of the sample...”. However, that estimation in Ref. 8 is for 45 μm thickness (Fig. S8 in Ref. 8), while the thickness used in this work is 80 μm , leading to a much larger temperature gradient. As the authors discuss, the current result will be able to be used to set the lower bound of HDL fraction. However, I believe the authors can better discuss this issue, together with a possible thickness inhomogeneity. Related to this, in their previous work, the temperature increase in HDL using the identical laser (2000 nm wavelength, 250 $\mu\text{J}/\text{pulse}$ with a spot size of 70 μm) was 90 K, compared to 60 K in this work. What caused this difference?

We agree that these issues merit clarification. To describe our method to estimate the temperature gradient in the sample we added the following text and figure in the Supplementary Information:

The temperature profile in a slab is simulated and shown in Figure S1(B). The IR laser spectrum at PAL-XFEL is assumed to have a Gaussian profile centered at 2 μm wavelength

and a bandwidth spread (2σ or FWHM) of $0.30\ \mu\text{m}$. The laser profile was convoluted with the absorption spectrum of ice and the average temperature jump is assumed to be $60\ \text{K}$. As shown in Figure S1(B), the temperature profile in the ice slab decreases by approximately $20\ \text{K}$ across the thickness of the sample (i.e. $60\ \text{K} \pm 10\ \text{K}$).

Figure S1. (B) Estimated temperature gradient in an ice slab (thickness= $60\ \mu\text{m}$) as a function of distance.

In the Methods section on “Temperature-Jump Estimation” we have added the sentence:

As shown in Supplementary Information, we estimate the variation of the temperature in a sample of thickness $60\ \mu\text{m}$ to be $60 \pm 10\ \text{K}$.

Regarding the temperature increase induced by the IR laser pulse, we have measured the final temperature in both this experiment and in Ref. 8 using the same procedure, based on the shift in the position of the ice Ih (110) Bragg peak as described in detail in the Methods section, and so both estimates should be of similar accuracy. The difference in the T-jump induced by the same laser pulse applied to LDA compared to HDA most likely arises from a difference in heat capacity between the two samples. There are several possibilities for the origin of this difference: (i) The heat capacity of LDL is higher than in HDL. (ii) We are starting the heating process at a higher temperature in the case of LDA and we expect that the heat capacity of both LDA and HDA increases with increasing temperature in this regime. (iii) If we are directly approaching the neighborhood of a critical point during the laser heating process in the LDA experiment, then this could result in a dramatic increase in the heat capacity not encountered when heating HDA. We agree that this difference when heating LDA and HDA merits further investigation, but without data to clarify the relative importance of the possible contributions, we have not included a discussion of this point in the manuscript.

The temporal evolution of HDL and LDL, as well as ice, is discussed using Fig. 4. Here, the guide to the eye is introduced smartly but is not necessarily justified in my view. From 8.4 ns to

~ 150 ns, the guide to the eye indicates a slight increase in the HDL fraction and a slight decrease in the LDL fraction, but they may be interpreted as constant within the uncertainty. The definition of uncertainty is not shown, and the number of XFEL shots used for obtaining single data points is not described either. This lack of description and justification significantly weakens the discussion from L.181 to L.216. For example, the statement “The gradual decrease in x_H occurring for $\Delta t > 300$ ns is consistent...” at line 214 is solely based on the result that a single data point at $\Delta t = 1 \mu\text{s}$ is slightly smaller than that at $\Delta t = 300$ ns before the formation of ice is observed. No gradual decrease can be discussed from the current dataset.

To address the Reviewer’s concerns regarding the interpretation of the data in Fig. 4, we have added an explanation of the error bars in the caption of Fig. 4:

Each data point is obtained by fitting our $\Delta I(q)$ curves, averaged over three x-ray shots, to a sum of contributions due to LDA, HDA and crystalline ice, using the same procedure as described in Ref. [8]. The error bars show the standard error for each data point determined by propagating the error associated with the averaged $\Delta I(q)$ curves.

We have also extended the range of the time axis in Fig. 4 (shown below) by an order of magnitude to $100 \mu\text{s}$ to better reveal the overall trends in all components.

The Reviewer correctly points out that the size of the error bars in Fig. 4 relative to the variation in the data does not permit the unambiguous identification of a maximum in the HDL curve prior to the onset of crystallization. We have modified the manuscript to acknowledge this; see the blue text in the revised manuscript on lines 203 to 210. Despite the uncertainty about the presence and location of a maximum in the HDL data, it remains the case that our data unambiguously demonstrate the existence of a HDL component in the

sample throughout the time scale prior to the onset of ice crystallization. This is our main result. In addition, the trend in the HDL data suggests weak initial growth of the HDL component, despite the fact that the error bars make firm conclusions impossible. We feel this trend merits comment given the correspondence of the time scale for this growth trend and the relaxation time of the LDL phase, and so we have retained this in the manuscript, with qualifications that acknowledge the uncertainties of the data points. We have removed the discussion related to the decreasing trend in the HDL component between 300 ns and the onset of ice formation at 3 μ s. This is a secondary point relative to our main findings, and as noted by the Reviewer, relates only to the behavior of a single data point at 1 μ s.

The authors exclude the possibility of the formation of nm-scale LDL-like & HDL-like domains, while their SAXS results indicate the appearance of nm-scale nucleation or spinodal decomposition process. Even though the authors describe the growth of macroscopic HDL and LDL observed in WAXS follows the appearance of SAXS intensity, in my view, they occur almost simultaneously.

We have modified the text describing the SAXS results to clarify the time scales on which nm-scale and macroscopic domains of LDL and HDL occur. We do not exclude the possibility of the formation of nm-scale domains. We state that our SAXS results (Fig. 5) show that they form at early times up to approximately 60 ns. However, on a time scale between 60 ns and several hundred ns, the SAXS intensity clearly decreases (Fig. 5b) but the WAXS intensity for HDL and HDL remains approximately constant (Fig. 4), consistent with the evolution of nm-scale domains into larger domains over time.

The revised text in the manuscript now reads:

The enhanced SAXS scattering, which is increasing for delay times up to approximately 60 ns, is consistent with the appearance of nm-scale LDL and HDL domains. These small domains may arise due to rapid nucleation at multiple sites throughout the sample, or a spinodal decomposition process. On a time scale between 60 ns and several hundred ns, the SAXS intensity decreases, while the intensity of the signal from our WAXS measurements remains approximately constant, as shown in Fig. 4, consistent with the consolidation and growth of macroscopic HDL and LDL domains from the nm-scale domains appearing at earlier times.

L.351-363: the authors discuss the scattering intensity from a two-component system. I understand the authors' intention here, but eqs.1 and 2 are not necessarily correct, particularly for the X-rays from XFEL. Instead of $(S_1+S_2)^2$, the intensity is the absolute square of summed scattering amplitude, so the cross-term can be almost cancelled even if the interface cannot be neglected. As I said, I understand the authors'

intent, but considering the readership of Nature Communications, this paragraph needs to be revised.

We agree this is a complex topic, which may not be appropriate to address in summary form here. Ref. 32 contains an extended discussion of the main points made in our manuscript in the Methods section “Distinguishing the LLPT from other scenarios”. Given this redundancy, and the Reviewer’s concern, we have simply removed this section from Methods and refer the reader to Ref. 32 in the main text on line 180.

Despite the several concerns raised above, I mostly support the authors’ argument, which needs more justifiable discussion. However, I am not sure this work deserves publication in Nature Communications, considering rather incremental progress compared to their previous work.

We note the expression of support by the Reviewer for our arguments and hope the clarification of the experimental procedures and analysis contained in the revised manuscript addresses the concerns expressed in this regard.

Regarding the concern that the present work is incremental compared to Ref. 8, we would like to emphasize the following:

- Ref. 8 produced evidence for the LLPT by heating HDA to a point well above (in pressure) the coexistence line of the LLPT, and then observing the phase transition as the sample then decompressed through the conditions of the LLPT. Here, evidence of the LLPT is observed by using a completely different starting sample, LDA, and via a much more direct path, which drives the sample directly into the coexistence region inside the LDL-HDL binodal. The procedure described here may therefore be more promising for future experiments attempting to directly access the conditions of the critical point of the LLPT.
- Our result that evidence for the LLPT can be found by heating LDA, rather than HDA, is, we believe, a surprising one. As we show, this is only possible because of the peculiar shape of the LDL-HDL binodal. Our work is the first to demonstrate the significance of this feature of the LLPT in supercooled water.
- Our results provide experimental evidence that the density, and therefore the pressure, of the critical point of the LLPT may be lower than most predictions suggest. This has important implications for choosing the conditions to explore in future experiments seeking evidence of the LLPT, and also for improving simulation models of water that will be accurate in the supercooled regime.

None of the above insights is anticipated by the results of Ref. 8. Indeed, Reviewer #1 emphasizes the significance of these points when recommending to publish our work.

REVIEWERS' COMMENTS

Reviewer #2 (Remarks to the Author):

The authors have addressed the comments in full detail, and the manuscript has been improved, in my view. I would recommend the paper for publication in Nature Communication in its present form.

The comments from Reviewer #2 are:

“The authors have addressed the comments in full detail, and the manuscript has been improved, in my view. I would recommend the paper for publication in Nature Communication in its present form.”

These comments do not request any changes to the manuscript and do not require a response.